# Effect of Flaxseed Supplementation on Milk and Plasma Fatty Acid Composition and Plasma Parameters of Holstein Dairy Cows

**DOI:** 10.3390/ani12151898

**Published:** 2022-07-26

**Authors:** Guoxin Huang, Jie Wang, Kaizhen Liu, Fengen Wang, Nan Zheng, Shengguo Zhao, Xueyin Qu, Jing Yu, Yangdong Zhang, Jiaqi Wang

**Affiliations:** 1Key Laboratory of Quality & Safety Control for Milk and Dairy Products of Ministry of Agriculture and Rural Affairs, Institute of Animal Sciences, Chinese Academy of Agricultural Sciences, Beijing 100193, China; huangguoxin1991@163.com (G.H.); wang.jie2@yuexiu.com (J.W.); 18728187930@163.com (K.L.); wfe8520382@163.com (F.W.); zhengnan_1980@126.com (N.Z.); zhaoshengguo1984@163.com (S.Z.); 2State Key Laboratory of Animal Nutrition, Institute of Animal Sciences, Chinese Academy of Agricultural Sciences, No. 2 Yuanmingyuan West Road, Haidian District, Beijing 100193, China; 3College of Animal Sciences and Technology, Northeast Agricultural University, Harbin 150030, China; 4China Excellent Milk Academy (Tianjin) Co., Ltd., Beichen District, Tianjin 300400, China; qu_xueyin@163.com (X.Q.); mengdejituan@126.com (J.Y.)

**Keywords:** flaxseed, α-linolenic acid, milk ALA/ALA intake, omega-3 polyunsaturated fatty acid

## Abstract

**Simple Summary:**

Few articles have reported the influence of whole and ground flaxseed intake on milk longer chain n-3 polyunsaturated fatty acid (PUFA) and the efficiency of diet α-linolenic acid (ALA) to convert to milk (milk ALA/ALA intake ratio). The objective of this experiment was to explore the effect of whole and ground flaxseed supplementation on the composition of n-3 polyunsaturated fatty acid (PUFA) in milk from dairy cows. This result indicated that whole and ground flaxseed supplementation can increase total n-3 PUFA. Compared with whole flaxseed, ground flaxseed supplementation showed higher content and yield of ALA in milk and milk ALA/ALA intake.

**Abstract:**

The objective of this study was to determine the effect of whole flaxseed and ground flaxseed supplementation on the composition of fatty acids in plasma and milk, particularly the content of omega-3 polyunsaturated fatty acids (n-3 PUFAs). Thirty Holstein dairy cows were randomly assigned to three treatment groups. Cows were fed a total mixed ration without flaxseed (CK), 1500 g of whole flaxseed (WF), and 1500 g of ground flaxseed (GF) supplementation. There were no differences observed in dry matter intake, milk yield, energy-corrected milk, and 4% fat-corrected milk (*p* > 0.05). Compared with the CK group, the contents of α-linolenic acid (ALA), eicosatrienoic acid, and eicosapentaenoic acid increased in the plasma and milk WF and GF groups, and the content of docosahexaenoic acid and total n-3 PUFA was higher in GF than the other groups (*p* < 0.001). The ALA yield increased to 232% and 360% in WF and GF, respectively, compared to the CK group. Compared with the WF group, GF supplementation resulted in an increased milk ALA/ALA intake ratio (*p* < 0.001). Flaxseed supplementation increased the activity of GSH-Px and decreased the concentration of MDA in milk (*p* < 0.001). Plasma parameters did not differ among the treatments (*p* > 0.05). This result indicated that compared with the WF group, GF supplementation in the diet showed higher efficiency in increasing the total n-3 PUFA levels and the milk ALA/ALA intake ratio, and decreased the ratio of n-6 PUFAs to n-3 PUFAs in milk.

## 1. Introduction

Nutrients such as omega-3 and omega-6 polyunsaturated fatty acids (n-3 and n-6 PUFAs, respectively) are beneficial for general human health [1,2]. However, an appropriate ratio of n-6 to n-3 PUFAs is more beneficial to human health [3]. Previous studies reported that a ratio of n-6 to n-3 PUFAs of 2:1 to 3:1 can suppress the progression of rheumatoid arthritis in humans [4]. In daily life, humans can obtain n-6 PUFAs more extensively than n-3 PUFAs, which causes an unhealthy ratio of n-6 to n-3 PUFAs of 20:1 or even higher in western diets [5]. The main sources of n-3 PUFAs are flaxseed and fish oil. Cow milk is a possible source of n-3 PUFAs, and it contains various n-3 PUFAs, including α-linolenic acid (ALA), eicosatrienoic acid (ETE), eicosapentaenoic acid (EPA), and docosahexaenoic acid (DHA).

The n-3 PUFA content in milk can be influenced by the composition of the diet, and therefore flaxseed is the main source of n-3 PUFAs in diets used for dairy cows. Previous studies have shown that flaxseed supplementation can increase the concentration of n-3 PUFAs, particularly ALA, in milk [6,7,8,9] and plasma [10]. Flaxseed is rich in ALA; 18% of the total seed (dry matter [DM] basis) or 53–56% of the total fatty acid content in flaxseed is composed of ALA [11,12]. Previous studies have shown that flaxseed can be used in many forms to increase the content of n-3 PUFAs in milk, including whole [13], ground [14], heat-treated [15], extruded [16], rumen-passing protected flaxseed [17], and flaxseed oil [18]. However, supplementation with the various flaxseed forms has shown different effects on ALA concentration in milk. Among the flaxseed forms, supplementation with mechanically treated flaxseed can increase the concentration of ALA in milk compared with whole flaxseed (WF), extruded flaxseed, and flaxseed oil, and no difference has been observed between rumen-passing protected and ground flaxseed (GF) treatments [19]. Therefore, GF may be suitable to improve the concentration of milk n-3 PUFAs. In addition, milk fatty acid profile is known to be a fingerprint of the health status of the cow [20,21,22], thus flaxseed in the diet can also influence the health of dairy cows.

Supplementation with GF can improve the concentration of ALA and linoleic acid (LA) [23] in milk. However, few studies have reported the influence of WF and GF intake on the content of long-chain n-3 PUFAs in milk, the ratio of n-6 to n-3 PUFAs in milk, and milk ALA/ALA intake. Thus, the aim of this study was to determine the effects of WF and GF supplementation in the diet on fatty acid composition in plasma and milk, milk production, and plasma parameters of dairy cows.

## 2. Materials and Methods

### 2.1. Cows and Experimental Design

The animal diet and experiment design were reported in our previous study [24]. Briefly, thirty Holstein dairy cows (days in milk: 90 ± 28 d; body weight: 628 ± 103 kg; milk yield: 37.22 ± 2.60 kg) were divided into three groups (10 cows per group) randomly, and the dietary treatments were a basal diet (CK) or supplemented with 1500 g of whole flaxseed (WF) or ground flaxseed (GF), as shown in supplementation Appendix A. The cows were fed individually, and had access to fresh water freely. The total mixed ration was fed daily at 06:30, 14:00, and 21:30 h to ensure 5~10% refusals. The GF was stored at room temperature (25~30 ℃) and used within a week to avoid oxidation. The cows were milked at 06:00, 13:30, and 21:00 h, and the milk yield was recorded individually.

### 2.2. Sampling, Measurements, and Analysis

Dry matter intake (DMI) was recorded daily for each cow. Samples of feed and orts from each treatment were collected twice per week and mixed together and stored at −20 °C for further analysis. The DM content was first analyzed by drying at 65 °C for 24 h, then 105 °C for 2 h to a constant weight. Crude protein (CP) content in the feed was measured using the Kjeldahl nitrogen test described by AOAC [25]. The neutral detergent fiber (NDF) and acid detergent fiber (ADF) contents of feed and orts were measured using heat-stable amylase (type XI-A of *Bacillus subtilis*; Sigma-Aldrich, St. Louis, MO, USA) [26]. The composition of fatty acids in the feed was analyzed using the method of Sukhija and Palmquist [27]. The cows were milked three times per day, and milk yield was recorded by the Milking equipment (Afimilk, Afikim, Israel). The milk samples were collected at the end of each week (5 times) according to the milking volume ratio of morning: afternoon: night = 4: 3: 3. Each milk sample was separated into three tubes, with one tube preserved with bronopol-B2 preservative (D&F Control System Inc., Dublin, ON, Canada) at 4 °C for milk fat, protein, lactose, nonfat milk solids, and total milk solid content analyses using a mid-infrared machine (Foss MilkoScan, Foss Food Technology Corp., Eden Prairie, MN, USA). The other two tubes were stored at −20 °C for further analysis. The energy-corrected milk (ECM) and 4% fat-corrected milk (4% FCM) were calculated using the equations described by Tyrrell and Reid (1965) [28]. On the last day of the experiment, approximately 30 mL of blood from the coccygeal vein was collected at 4 h after feeding [29]. Plasma was obtained after centrifuging the blood at 3000× *g* for 15 min at 4 °C. Each plasma sample was divided into three serum separator tubes and stored at −20 °C for further analysis. The profile of plasma and milk fatty acids were analyzed using the method described in Wang et al. [30]. The activities of glutathione peroxidase (GSH-Px), catalase (CAT), Total antioxidant capacity (T-AOC), and the concentration of malondialdehyde (MDA) in plasma and milk samples were determined using kits according to the manufacturer’s instructions (Nanjing Jiancheng Bioengineer Institute, China). Triglyceride (TG), high-density lipoprotein cholesterol (HDLC), low-density lipoprotein cholesterol (LDLC), very low-density lipoprotein cholesterol (VLDLC), total cholesterol (TC), Aspartate aminotransferase (AST), Alanine aminotransferase (ALT), alkaline phosphatase (ALP), total protein (TP), albumin (ALB), globulin (GLB), blood urea nitrogen (BUN), and glucose (GLU) levels in plasma were analyzed using an automatic plasma analyzer (Mindray BS200, Shenzhen, China).

### 2.3. Statistical Analysis

The DMI, milk yield, 4% FCM, ECM, ECM/DMI, milk composition, and milk production were determined using MIXED models of SAS (version 9.4, SAS Inst., Inc., Cary, NC, USA). Fixed effects included treatment, week, and the treatment × week interaction; random effects included cows nested within treatments. The milk and plasma fatty acid composition, parameters and antioxidant indexes were analyzed using ANOVA models of SAS. Significant and extremely significant levels were set at *p* ≤ 0.05 and *p* ≤ 0.01, respectively. Correlation analysis between the n-3 and n-6 PUFA levels in milk and plasma, the antioxidant indexes, and the n-3 and n-6 PUFA in milk were performed using Spearman’s rank correlation. The significance level was *p* ≤ 0.05 and the extreme significance level was *p* ≤ 0.01.

## 3. Results

### 3.1. Feed Fatty Acid Composition

The content and composition of fatty acids in the diets were influenced by flaxseed supplementation (Table 1). In this study, ALA content was higher in the WF and GF than in the CK diet (20.60% and 19.80% vs. 2.10%, respectively). The LA concentration was lower in the WF and GF than in the CK diet (9.84% and 9.49% vs. 18.86%, respectively).

### 3.2. Feed Intake, Milk Yield, and Milk Composition

The effects of WF or GF supplementation on feed intake, milk yield, and milk composition are shown in Table 2. There were no significant differences among treatments for DMI (*p* = 0.756), milk yield (*p* = 0.122), 4% FCM (*p* = 0.658), ECM (*p* = 0.524), and gross feed efficiency (ECM/DMI) (*p* = 0.187). The WF supplementation increased the content of milk protein (*p* = 0.037) compared to the CK group, but there was no difference between WF and GF groups. The other components of milk composition and milk production did not differ among the three groups.

### 3.3. Plasma Fatty Acid Composition

Compared with the CK group, supplementing flaxseed had no effect on the concentration of fatty acids with a carbon number lower than 16 (*p* > 0.05), and decreased the content of C16:1 trans-7 (*p* < 0.001; Table 3). Flaxseed supplementation had no effect on Conjugated linoleic acid (CLA), including C18:2 cis-9,trans-11 and C18:2 trans-10,cis-12, but increased the content of ALA in plasma, which was higher in the GF than the WF group (*p* < 0.001). The WF and GF supplementation increased the concentration of plasma ALA to 221% and 275%, respectively. For longer n-3 PUFA, the content of plasma EPA was higher in the WF and GF groups compared to CK, with no difference between WF and GF. The total n-3 PUFA significantly differed following flaxseed supplementation and was highest in the GF group (*p* < 0.001). The GF supplementation reduced the n-6 PUFA content in the plasma (*p* < 0.001), but there was no difference between the CK and WF groups. The ratio of n-6 to n-3 PUFA decreased with flaxseed supplementation and was lower in the GF than in the WF group.

### 3.4. Milk Fatty Acid Composition

The WF and GF supplementation decreased the concentration in milk of fatty acids with fewer than 18 carbons (*p* < 0.05), and increased those with 18 carbons, including total C18:0 (*p* = 0.016), C18:2 cis-9,trans-11 (*p* < 0.001), C18:2 cis-9,cis-12 (LA) (*p* = 0.020), and ALA (*p* < 0.001; Table 4). As expected, flaxseed supplementation increased the milk ALA content (*p* < 0.001). In the WF and GF groups, the milk ALA concentration increased (223% and 350%, respectively) as did yield (232% and 360%, respectively), compared with the CK group (Table 5). Compared to WF, the GF group had higher milk ALA content (0.91 vs. 0.58 g/100 g fatty acid) and yield (13.51 vs. 8.71 g/d). For longer n-3 PUFA, flaxseed supplementation increased the content or yield of ETE and EPA, but with no difference between the WF and GF groups. Milk DHA content only differed between the CK and GF groups, with a higher yield in the GF group. The ratio of n-6 to n-3 PUFA also decreased in milk with flaxseed supplementation and was lowest in the GF group. Compared with CK, the yield of total n-3 fatty acids increased in the WF and GF groups, with the highest in the GF group. Increasing ALA intake reduced the ratio of milk ALA/ALA intake. Compared with the WF group, feeding GF increased milk ALA/ALA intake.

### 3.5. Antioxidant Indexes in Plasma and Milk

In plasma, flaxseed supplementation in a diet showed no effect on the activity of GSH-Px and the concentration of MDA (*p* > 0.05) (Table 6). The activity of CAT in cows receiving flaxseed (WF and GF groups) was higher than in the CK group (*p* < 0.01). T-AOC in the WF group was lower than in the CK group (*p* < 0.01); however, no significant different showed between the CK and GF groups. In milk, flaxseed supplementation in the diet showed no effect on the activity of CAT and T-AOC (*p* > 0.05). The concentration of MDA was decreased in the flaxseed treatment group (WF and GF groups) compared with the CK group (*p* < 0.01). Moreover, the activity of GSH-Px was higher in the WF and GF groups compared with the CK group (*p* < 0.01).

### 3.6. Correlation Analysis

Most of the n-3 PUFA concentrations showed a positive correlation in plasma and milk (Figure 1). Milk ALA, ETE, EPA, and total n-3 PUFA levels showed positive correlations with ALA, ETE, EPA, and total n-3 PUFA levels in plasma, respectively. However, the correlation between milk DHA and plasma DHA levels (*p* < 0.01) had a higher level of significance than plasma ALA, ETE, and EPA levels (*p* < 0.05). Most of the n-3 PUFA concentrations in milk showed a negative correlation with n-6 PUFA in plasma.

### 3.7. Plasma Parameters

We determined 13 plasma indicators in this experiment: TG, LDLC, VLDLC, HDLC, TC, AST, ALT, ALP, TP, ALB, GLB, BUN, and GLU (Table 7). There were no differences in the plasma parameters among the treatments (*p* > 0.05).

## 4. Discussion

### 4.1. Feed Intake and Milk Yield

A previous study showed that an increase in dietary fat may cause a decrease in DMI in dairy cows [31], especially unsaturated fatty acids [32]. Flaxseeds are rich in unsaturated fatty acids. Resende et al. [33] reported that the DMI decreased linearly with an increased amount of GF supplementation. In the current study, GF and WF supplementation at 1500 g had no effect on DMI. This result was similar to that reported by Martin et al. [34]. Do Prado et al. [35] also reported that supplementation with 4.8% WF in the diet had no effect on DMI. Isenberg et al. [14] found that supplementation with 10% GF had no effect on DMI. In a meta-analysis, Leduc et al. [19] found that WF and GF supplementation in the diet had no influence on DMI. Thus, the amounts may be the main factor that regulated the DMI of dairy cows in our flaxseed supplementation experiment, and the addition of GF and WF was not sufficient to cause a decrease in DMI. The effect of oilseed supplementation on milk yield varies. Brzozowska et al. [6] found that WF supplementation could increase milk yield, and Hurtaud et al. [36] reported that milk yield increased with flaxseed supplementation. However, some studies reported that flaxseed supplementation does not influence milk yield. Martin et al. [34] reported no influence on milk yield after supplementation with flaxseed at 12.4% of the diet DM. Caroprese et al. [37] also found that supplementation with WF at 6.5% of the diet DM had no effect on milk yield. Similar findings were also reported by Caroprese et al. [38]. This difference may be due to flaxseeds being rich in oil with high energy that can balance the energy in the diet. However, this balance correlated highly with the type of diet. Therefore, the effect of flaxseed supplementation on milk yield was variable. In the current study, GF and WF supplementation showed no effect on milk yield. Moreover, no difference was observed between treatments in gross feed efficiency (ECM/DMI). This result was similar to that of Resende et al. [33], who fed cows with GF at 5%, 10%, and 15% of the diet DM. In general, our WF and GF supplementation had no effect on DMI, milk yield, 4% FCM, ECM, and ECM/DMI.

### 4.2. Milk Composition

Previous studies showed that dietary oil supplements can reduce milk fat content [32]. Bu et al. [29] reported that supplementation with 4% flaxseed oil in DM diet could decrease milk fat content. Furthermore, supplementation with extruded flaxseeds also decreased milk fat content [39,40]. However, different forms of flaxseed showed a different release pattern of linseed FA [37], and milk fat depression may be influenced by the form of flaxseed supplementation. Caroprese et al. [37] found that WF supplementation positively affected milk fat percentage. In our experiment, supplying WF or GF at 1500 g per day had no effect on the content and yield of milk fat. This result was similar to that of Petit et al. [41], who reported that milk fat content and yield did not differ between the control and the 9.7% WF group. The same result was also observed by Do Prado et al. [35]. Stergiadis et al. [42] observed no effect on milk fat content in conventional or organic cows fed a diet with 1.5 kg of rolled flaxseed. Milk protein content can be influenced by dietary fat [43] and lactation. However, the forms of flaxseed may influence the content of protein in milk. Previous research reported that WF supplementation in the diet of dairy cows can increase the content of protein in milk [37]. However, a study by Isenberg et al. [14] showed no influence on the content of milk protein with GF supplementation in the diet. These results were similar to those of the present study. In addition, the present study found no difference between GF and WF groups, and a similar result was also found in a study by Oba et al. [44]. This may be due to the fact that, compared with GF, WF showed higher efficiency in decreasing the degradability of the protein in the rumen, and therefore increased protein in milk.

### 4.3. Plasma Fatty Acid Composition

Flaxseed is rich in ALA [45], which is a C18 fatty acid. Thus, flaxseed supplementation did not affect the concentration of fatty acids with a carbon number lower than 16. He et al. [46] reported that 14.13% GF supplementation had no effect on the concentration of 14:0 and 14:1 cis-9 and decreased the content of C16:1 cis-9 in plasma, which was similar to our study. A previous study showed that most of the long-chain fatty acids in plasma originate from the diet [47]. Previous results [10] and our research showed that the ALA content in plasma increased with dietary ALA intake. The ALA in GF and WF were released in different ways. The GF can release the ALA directly into the rumen; however, WF needs to be crushed by chewing, and then releases the ALA [24]. Thus, some WF is directly excreted out of the body [44]. In this experiment, there was a higher concentration of ALA in the GF than in the WF group. Both EPA and DHA can be synthesized from ALA in the body [1]. In this experiment, flaxseed supplementation also increased the EPA concentration in plasma. Although DHA can be synthesized in the body using ALA, the amounts are extremely limited [48]. In this experiment, no difference was observed in plasma DHA content between the treatments. Flaxseed supplementation can reduce the n-6 PUFA content in plasma [49]. In our study, a similar result was only found for GF supplementation. Flaxseed supplementation decreased the n-6 to n-3 PUFA ratio, with a lower ratio in the GF than in the WF group.

### 4.4. Milk Fatty Acid Composition

Oil seed supplementation can influence the composition of milk fatty acids [50]. Brzozowska et al. [6] and Ward et al. [51] reported that WF supplemented in the diet of cows could decrease the fatty acids with a number of carbons lower than 18 in milk. Akraim et al. [52] reported that supplementation with 16.42% GF also produced a similar result. Furthermore, GF supplementation decreased the content of C16:0 in conventional and organic cow milk [42]. In the present study, flaxseed supplementation decreased the concentration of most fatty acids with a number of carbons lower than 18 in milk than the CK group, and there was no difference between the WF and GF groups. Flaxseed supplementation in the diet can be used to increase the concentration of ALA in milk [53]. The ALA can be converted by biohydrogenation to C18:2 cis-9,trans-11 in the rumen [54] and increase the content of C18:2 cis-9,trans-11 in milk [33]. In the present study, flaxseed supplementation increased the concentration of ALA and C18:2 cis-9,trans-11 in milk, with a higher concentration in the GF than in the WF group. Longer chain n-3 PUFAs can be synthesized from ALA through metabolic pathways in vivo [1]. Flaxseed supplementation can also influence the ETE, EPA, and DHA contents in milk. Petit et al. [41] reported that the EPA content increased with 9.7% WF supplementation (0.56% and 0.1% for the experiment and control groups, respectively). The same result was also observed by Resende et al. [33]. However, some studies also found flaxseed supplementation had no effect on EPA content in milk [55] or decreased it [42]. Some studies found that flaxseed supplementation does not affect milk DHA [35,37]. In the current study, flaxseed supplementation increased the content or yield of ETE and EPA, but with no difference between WF and GF groups. ETE was the intermediate product of ALA-synthesized EPA and DHA [1], which can increase with ALA supplementation in the diet. ETE, as a kind of n-3 PUFAs, may play an important role in vivo, but there are few studies focused on it. Moreover, the DHA content was higher in the GF group compared to CK, but no different between the WF and CK groups. Milk fatty acids are transferred from mammary gland cells, and DHA might be synthesized from ALA in these cells, but in limited amounts [43]. Thus, a higher ALA content increases the DHA content in milk. Flaxseed supplementation increased the content of total n-6 PUFAs and total n-3 PUFAs in this experiment. The ratio of n-6 to n-3 PUFAs also decreased in milk with flaxseed supplementation, consistent with the study of Moallem et al. [53] and was lower in the GF than in the WF group of the present study.

The n-3 PUFAs are transferred from the plasma and may be synthesized from ALA in mammary gland cells. Thus, plasma ALA showed positive correlations with ALA, ETE, EPA, DHA, and total n-3 PUFAs in milk. However, the conversion of DHA in tissues is limited [56]. The DHA can be converted into EPA in the body [57], so it likely resulted in a positive correlation between plasma DHA and milk EPA. LA and ALA share the same family of enzymes in the formation of higher unsaturated fatty acids [58]. Therefore, there was a negative correlation between milk total n-3 PUFA and plasma total n-6 PUFA levels. In this study, we also found that most plasma n-6 PUFAs showed a negative correlation with n-3 PUFAs in milk.

### 4.5. Antioxidant Indexes in Plasma and Milk

As we know, n-3 PUFA showed high antioxidant activity [59], and a diet with n-3 PUFA can increase the activity of antioxidants in blood [60]. CAT is one of these antioxidants, and in this study, flaxseed supplementation can increase the activity of CAT in plasma. GSH-Px is also an important antioxidant index in the blood, and it was previously reported that the increase in diet of n-3 PUFAs could increase the activity of GSH-Px in plasma [60]. This study showed that dietary supplementation of whole and ground flaxseed increased the activity of GSH-Px in plasma but did not differ from that of the control group.

The diet with n-3 PUFAs can influence the capacity of antioxidants in milk, which influences the health of the infant animals. Prescott et al. [61] found that pregnant women’s consumption of rich n-3 PUFAs may reduce oxidative stress in infants. Similar results were found in this study, and supplementation with whole and ground flaxseed contributed to the activity of GSH-Px and reduced the content of MDA in milk. This can reduce oxidative stress in infant animals. Moreover, the level of milk ALA, ETE, EPA, and total n-3 PUFA were positively correlated with GSH-Px and negatively correlated with MDA.

### 4.6. Plasma Parameters

Blood has the functions of transporting nutrients, regulating body fluid balance, maintaining internal environment stability, and participating in immunity. Plasma parameters are closely related to body metabolism and can reflect animal health. Previous studies found that the blood fatty acid can influence the health of dairy cows [62,63]. As we know, TG, HDLC, LDLC, VLDLC, and TC are important components of plasma lipids. The LDLC levels are correlated with cardiovascular diseases [64]. These components can be influenced by dietary fat [65]. Previous studies reported that flaxseed oil supplementation can increase the concentration of TC and LDLC in plasma [29,66]. The increase in TC and LDLC concentrations in plasma is detrimental to animal cardiovascular health. However, no effects were observed in the present study. High-fat diets can cause fatty liver hemorrhagic syndrome in animals [67]. Plasma AST and ALP are used as indicators to reflect liver damage [68], and high levels lead to liver damage [69]. Previous studies reported that ALP showed a positive correlation with PUFA in milk [20]. The GLB is an important immunological active protein in the plasma [70]. In the present study, flaxseed supplementation also showed no effects on plasma GLB, AST, and ALP. Thus, GF or WF supplementation could increase dietary fat levels, but in this experiment, it did not harm the cardiovascular and liver health of dairy cattle.

Flaxseed supplementation in the diet can increase the content of ALA, ETE, EPA, and total n-3 PUFAs and decrease the n-6 to n-3 PUFA ratio in milk and blood but showed no influence on the production performance and health of dairy cows. Due to the high antioxidant activity of n-3 PUFAs, a higher capacity of GSH-Px and lower MDA was found in WF and GF milk, compared with the CK group.

## 5. Conclusions

Supplementation of GF at 1500 g per day showed no effect on milk yield, 4% FCM, ECM, and ECM/DMI, and no effect on the health of dairy cattle. The increase in ALA, ETE, EPA, and total n-3 PUFAs and the decreased n-6 to n-3 PUFA ratio in milk were greater than in the CK and WF groups. Additionally, supplementing the cows’ diets with GF showed higher efficiency of milk ALA/ALA intake compared to WF.

## Figures and Tables

**Figure 1 animals-12-01898-f001:**
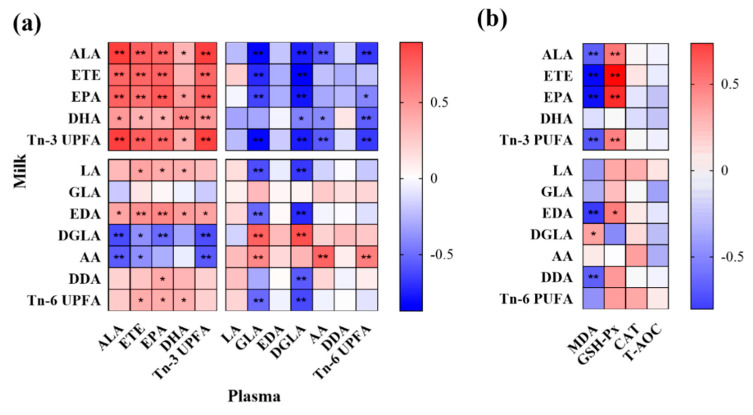
(**a**) Correlation analysis between the n-3 PUFA and n-6 PUFA levels in milk and plasma; (**b**) the antioxidant indexes and the n-3 PUFA and n-6 PUFA in milk. Red represents a positive correlation and blue represents a negative correlation. Significant correlations: * *p* ≤ 0.05, ** *p* ≤ 0.01.

**Table 1 animals-12-01898-t001:** Content and composition of fatty acid in experimental diets of different treatment groups.

Item	Treatments		
	CK	WF	GF
Total FA, % diet DM	4.59	8.52	8.50
FA ^1^, % of total FA reported			
C14:0	2.04	1.43	1.46
C16:0	76.05	67.02	68.04
C16:1 cis-9	0.03	0.03	0.04
C18:0	0.03	0.04	0.05
C18:1 cis-9	0.77	0.94	1.01
C18:2 cis-9,cis-12 (LA)	18.86	9.84	9.49
C18:3 cis-9,cis-12,cis-15 (ALA)	2.10	20.60	19.80
C20:0	0.04	0.03	0.04
C20:1 cis-11	0.01	0.01	0.01
C22:0	0.04	0.03	0.04
C24:0	0.03	0.02	0.03

CK = dairy cows fed a basal diet (without flaxseed); WF = dairy cows with whole flaxseed diet (whole flaxseed 1500 g per day); GF = dairy cows with ground flaxseed diet (ground flaxseed 1500 g per day); FA: Fatty acid; DM: Dry matter; ^1^ Expressed as number of carbons: number of double bonds.

**Table 2 animals-12-01898-t002:** Dry matter intake, milk composition, and milk yield of different treatment groups.

Item	Treatments			SEM	*p*-Value
	CK	WF	GF		
DMI, kg/d	22.47	21.89	22.36	0.330	0.756
Milk yield, kg/d	36.05	36.39	38.09	0.486	0.122
4% FCM ^1^, kg/d	36.78	37.05	37.96	0.537	0.658
ECM ^2^, kg/d	39.14	39.96	40.71	0.551	0.524
Efficiency, ECM/DMI	1.61	1.68	1.73	0.026	0.187
Milk composition, %					
Milk fat	4.15	4.12	3.95	0.082	0.583
Milk Protein	3.10 ^b^	3.31 ^a^	3.14 ^a,b^	0.035	0.037
Milk lactose	4.84	4.78	4.81	0.020	0.566
Nonfat milk solids	8.43	8.65	8.53	0.038	0.060
Total milk solids	12.58	12.77	12.48	0.092	0.451
Milk production, kg/d					
Milk fat yield	1.49	1.50	1.51	0.030	0.971
Milk protein yield	1.12	1.20	1.20	0.017	0.059
Milk lactose yield	1.74	1.74	1.84	0.025	0.168
Nonfat milk solids yield	3.04	3.15	3.27	0.043	0.087
Total milk solids yield	4.53	4.65	1.78	0.059	0.238

CK = dairy cows fed a basal diet (without flaxseed); WF = dairy cows with whole flaxseed diet (whole flaxseed 1500 g per day); GF = dairy cows with ground flaxseed diet (ground flaxseed 1500 g per day); DMI, dry matter intake; ECM, energy-corrected milk; 4% FCM, 4% fat-corrected milk; ^1^ 4% FCM = 0.4 × milk (kg) + 15 × fat (kg) (Tyrrell and Reid, 1965); ^2^ ECM = 0.327 × milk (kg) + 12.95 × fat (kg) + 7.20 × protein (kg); (Tyrrell and Reid, 1965); ^a,b^ Means in the same row with different superscripts differ significantly for treatment effect.

**Table 3 animals-12-01898-t003:** Plasma fatty acid composition of cows in different treatment groups.

Fatty Acid, % of Total	Treatments			SEM	*p*-Value
	CK	WF	GF		
C4:0	0.051	0.051	0.048	0.002	0.697
C6:0	0.021	0.023	0.022	0.001	0.722
C8:0	0.019	0.02	0.02	0.001	0.737
C10:0	0.035	0.041	0.039	0.002	0.439
C12:0	0.084	0.087	0.082	0.003	0.813
C14:0	0.56 ^a,b^	0.63 ^a^	0.52 ^b^	0.020	0.064
C14:1 cis-9	0.054	0.061	0.052	0.002	0.189
C16:0	10.16 ^a^	9.25^b^	9.86 ^a^	0.136	0.015
C16:1 trans-7	0.43 ^a^	0.36^b^	0.27^c^	0.014	<0.001
C16:1 cis-9	0.66	0.76	0.65	0.023	0.116
C18:0	12.33 ^a^	10.90 ^b^	12.17 ^a^	0.234	0.017
C18:1 cis-6	0.053 ^a^	0.048 ^a,b^	0.045 ^b^	0.001	0.037
C18:1 cis-9	4.7	4.88	4.48	0.106	0.318
C18:1 trans-9	0.36	0.34	0.27	0.019	0.105
C18:1 trans-6 + trans-11	0.62 ^b^	0.59 ^b^	0.77 ^a^	0.022	<0.001
other C18:1 ^1^	0.83 ^b^	0.81 ^b^	1.68 ^a^	0.077	<0.001
C18:2 cis-9,cis-12 (LA)	50.41 ^a,b^	51.82 ^a^	49.14 ^b^	0.360	0.006
C18:2 trans-9,trans-12	0.10 ^b^	0.09 ^b^	0.11 ^a^	0.002	<0.001
C18:2 cis-9,trans-11 (CLA)	0.20	0.20	0.21	0.005	0.483
C18:2 trans-10,cis-12 (CLA)	0.09	0.08	0.08	0.002	0.415
C18:3 cis-6,cis-9,cis-12 (GLA)	1.16 ^a^	0.87 ^b^	0.54 ^c^	0.058	<0.001
C18:3 cis-9,cis-12,cis-15 (ALA)	2.40 ^c^	5.30 ^b^	6.59 ^a^	0.341	<0.001
C20:0	0.12	0.10	0.10	0.002	0.080
C20:1 cis-9	0.05 ^a^	0.04 ^b^	0.04 ^b^	0.001	0.006
C20:1 cis-11	0.13 ^a^	0.09 ^b^	0.10 ^b^	0.005	<0.001
C20:2 cis-11,cis-14 (EDA)	0.099 ^a^	0.086 ^b^	0.0910 ^a,b^	0.002	0.017
C20:3 cis-8,cis-11,cis-14 (DGLA)	2.51 ^a^	1.61 ^b^	1.47 ^b^	0.103	<0.001
C20:3 cis-11,cis-14,cis-17 (ETE)	0.00 ^b^	0.05 ^a^	0.05 ^a^	0.004	<0.001
C20:4 cis-5,cis-8,cis-11,cis-14 (AA)	1.79 ^a^	1.71 ^a^	1.43 ^b^	0.058	0.022
C20:5 cis-5,cis-8,cis-11,cis-14,cis-17 (EPA)	0.31 ^b^	0.50 ^a^	0.56 ^a^	0.027	<0.001
C22:0	0.20 ^a^	0.17 ^b^	0.19 ^a^	0.004	0.007
C22:1 cis-13	6.49	5.75	5.51	0.200	0.112
C22:2 cis-13,cis-16 (DDA)	0.20 ^a^	0.18 ^b^	0.19 ^a,b^	0.005	0.078
C22:6 cis-4,cis-7,cis-10,cis-13,cis-16,cis-19 (DHA)	0.06	0.06	0.07	0.002	0.219
C24:0	0.25	0.23	0.25	0.005	0.135
C24:1cis-15	0.19 ^a^	0.16 ^b^	0.17 ^a,b^	0.004	0.075
Total n-6 PUFA ^2^	56.17 ^a^	56.27 ^a^	52.87 ^b^	0.409	<0.001
Total n-3 PUFA ^3^	2.76 ^c^	5.90 ^b^	7.26 ^a^	0.366	<0.001
n-6/n-3 ^4^	20.39 ^a^	9.67 ^b^	7.35 ^c^	1.075	<0.001
SFA	23.83 ^a^	21.49 ^b^	23.29 ^a^	0.271	<0.001
MUFA	14.70	14.23	13.96	0.234	0.440
PUFA	59.32 ^b^	62.54 ^a^	60.53 ^b^	0.380	0.001

CK = dairy cows fed a basal diet (without flaxseed); WF = dairy cows with whole flaxseed diet (whole flaxseed 1500 g per day); GF = dairy cows with ground flaxseed diet (ground flaxseed 1500 g per day); ^1^ Sum of unrecognized C18:1 in this experiment, ^2^ Sum of LA, GLA, EDA, DGLA, AA, and DDA; ^3^ Sum of ALA, ETE, EPA, and DHA; ^4^ The ratio of n-6 to n-3 PUFA; ^a–c^ Means in the same row with different superscripts differ significantly for treatment effect.

**Table 4 animals-12-01898-t004:** Milk fatty acid composition of cows in different treatment groups.

Fatty Acid, % of Total	Treatments			SEM	*p*-Value
	CK	WF	GF		
C4:0	3.23 ^a^	2.84 ^a,b^	2.67 ^b^	0.095	0.042
C6:0	2.40 ^a^	2.00 ^b^	1.91 ^b^	0.069	0.005
C8:0	1.50 ^a^	1.23 ^b^	1.20 ^b^	0.045	0.007
C10:0	3.08 ^a^	2.52 ^b^	2.47 ^b^	0.095	0.009
C10:1 cis-9	0.65 ^a^	0.53 ^b^	0.45 ^b^	0.024	0.001
C12:0	3.18 ^a^	2.58 ^b^	2.69 ^b^	0.091	0.012
C12:1 cis-9	0.09 ^a^	0.08 ^b^	0.07 ^b^	0.003	0.005
C14:0	9.85	9.00	9.42	0.168	0.119
C14:1 cis-9	0.89 ^a^	0.74 ^b^	0.72 ^b^	0.030	0.042
C16:0	35.07 ^a^	31.15 ^b^	30.55 ^b^	0.468	<0.001
C16:1 trans-7	0.13 ^b^	0.15 ^a^	0.12 ^b^	0.003	0.010
C16:1 cis-9	1.36	1.40	1.37	0.042	0.919
C18:0	10.15 ^b^	11.58 ^a^	11.27 ^a^	0.221	0.016
C18:1 cis-6	0.05 ^b^	0.07 ^a^	0.06 ^a,b^	0.003	0.070
C18:1 cis-9	18.36 ^b^	22.42 ^a^	19.67 ^b^	0.506	0.001
C18:1 trans-9	0.26 ^c^	0.32 ^b^	0.55 ^a^	0.024	<0.001
C18:1 trans-6 + trans-11	1.19 ^c^	1.53 ^b^	2.28 ^a^	0.101	<0.001
Other C18:1 ^1^	1.49 ^c^	2.21 ^b^	4.07 ^a^	0.206	<0.001
C18:2 cis-9,cis-12 (LA)	2.62 ^b^	2.80 ^a,b^	3.02 ^a^	0.060	0.020
C18:2 trans-9,trans-12	0.12 ^c^	0.18 ^b^	0.26 ^a^	0.011	<0.001
C18:2 cis-9,trans-11 (CLA)	0.46 ^c^	0.62 ^b^	0.83 ^a^	0.036	<0.001
C18:2 trans-10,cis-12 (CLA)	0.04	0.05	0.05	0.002	0.142
C18:3 cis-6,cis-9,cis-12 (GLA)	0.05	0.05	0.05	0.001	0.518
C18:3 cis-9,cis-12,cis-15 (ALA)	0.26 ^c^	0.58 ^b^	0.91 ^a^	0.052	<0.001
C20:0	0.10 ^b^	0.12 ^a^	0.11 ^a,b^	0.002	0.007
C20:1 cis-9	0.08 ^b^	0.09 ^a^	0.08 ^b^	0.002	0.019
C20:1 cis-11	0.033 ^b^	0.041 ^a^	0.042 ^a^	0.001	0.006
C20:2 cis-11,cis-14 (EDA)	0.035 ^b^	0.043 ^a^	0.044 ^a^	0.001	0.014
C20:3 cis-8,cis-11,cis-14 (DGLA)	0.012 ^a^	0.010 ^a,b^	0.009 ^b^	0.004	0.005
C20:3 cis-11,cis-14,cis-17 (ETE)	0.00 ^b^	0.03 ^a^	0.03 ^a^	0.002	<0.001
C20:4 cis-5,cis-8,cis-11,cis-14 (AA)	0.14 ^a^	0.13 ^a^	0.11 ^b^	0.003	<0.001
C20:5 cis-5,cis-8,cis-11,cis-14,cis-17 (EPA)	0.03 ^b^	0.05 ^a^	0.06 ^a^	0.002	<0.001
C22:0	0.05	0.06	0.06	0.002	0.164
C22:1 cis-13	0.08 ^b^	0.09 ^a,b^	0.10 ^a^	0.005	0.094
C22:2 cis-13,cis-16 (DDA)	0.03	0.03	0.03	0.001	0.278
C22:6 cis-4,cis-7,cis-10,cis-13,cis-16,cis-19 (DHA)	0.007 ^b^	0.009 ^a,b^	0.017 ^a^	0.002	0.037
C24:0	0.05	0.06	0.06	0.002	0.383
C24:1cis-15	0.02	0.02	0.02	0.001	0.357
Total n-6 PUFA ^2^	2.98 ^b^	3.16 ^a^	3.34 ^a^	0.059	0.042
Total n-3 PUFA ^3^	0.30 ^c^	0.67 ^b^	1.01 ^a^	0.057	<0.001
n-6/n-3 ^4^	10.07 ^a^	4.77 ^b^	3.30 ^c^	0.545	<0.001
SFA	68.66 ^a^	63.08 ^b^	62.45 ^b^	0.685	<0.001
MUFA	23.31 ^b^	27.61 ^a^	25.66 ^a^	0.540	0.002
PUFA	3.90 ^c^	4.67 ^b^	5.49 ^a^	0.141	<0.001
D9- desaturase indices					
C10:1	17.40 ^a^	17.44 ^a^	15.15 ^b^	0.376	0.012
C12:1	2.74	2.84	2.55	0.059	0.131
C14:1	8.35	7.61	7.10	0.297	0.230
C16:1	3.72	4.28	4.32	0.137	0.131
C18:1	64.34	65.86	63.51	0.552	0.218
C20:1	44.28	43.53	42.33	0.459	0.223

CK = dairy cows fed a basal diet (without flaxseed); WF = dairy cows with whole flaxseed diet (whole flaxseed 1500 g per day); GF = dairy cows with ground flaxseed diet (ground flaxseed 1500 g per day); ^1^ Sum of unrecognized C18:1 in this experiment; ^2^ Sum of LA, GLA, EDA, DGLA, AA, and DDA; ^3^ Sum of ALA, ETE, EPA, and DHA; ^4^ The ratio of n-6 to n-3 PUFA; ^a–c^ Means in the same row with different superscripts differ significantly for treatment effect.

**Table 5 animals-12-01898-t005:** Milk ALA, EPA, DHA, and total n-3 PUFA yield of different treatment groups.

Items	Treatments			SEM	*p*-Value
	CK	WF	GF		
Milk ALA yield, g/d	3.75 ^c^	8.71 ^b^	13.51 ^a^	0.791	<0.001
Milk ETE yield, g/d	0.00 ^b^	0.38 ^a^	0.38 ^a^	0.035	<0.001
Milk EPA yield, g/d	0.49 ^b^	0.77 ^a^	0.82 ^a^	0.029	<0.001
Milk DHA yield, g/d	0.09 ^b^	0.14 ^b^	0.25 ^a^	0.025	0.021
n-3 PUFA ^1^ yield, g/d	4.34 ^c^	10.00 ^b^	14.96 ^a^	0.855	<0.001
Milk ALA/ALA intake, %	17.30 ^a^	2.29 ^c^	3.61 ^b^	1.277	<0.001

CK = dairy cows fed a basal diet (without flaxseed); WF = dairy cows with whole flaxseed diet (whole flaxseed 1500 g per day); GF = dairy cows with ground flaxseed diet (ground flaxseed 1500 g per day); ^1^ Sum of ALA, ETE, EPA, and DHA; ^a–c^ Means in the same row with different superscripts differ significantly for treatment effect.

**Table 6 animals-12-01898-t006:** Antioxidant indexes in plasma and milk.

Item	Antioxidant Indexes	Treatments			SEM	*p*-Value
		CK	WF	GF		
Plasma	GSH-Px (ng/L)	40.99	42.95	42.99	0.606	0.823
	MDA (nmol/mL)	16.76	16.94	17.27	1.467	0.946
	CAT (ng/L)	35.63 ^b^	47.25 ^a^	48.08 ^a^	1.607	0.000
	T-AOC (nmol/mL)	1.49 ^a^	0.99 ^b^	1.28 ^a^	0.064	0.002
Milk	GSH-Px (ng/L)	273.31 ^c^	501.27 ^a^	433.47 ^b^	23.340	0.000
	MDA (nmol/mL)	6.51 ^a^	2.19 ^c^	3.81 ^b^	0.417	0.000
	CAT (ng/L)	5.82	7.03	5.09	0.633	0.465
	T-AOC (nmol/mL)	0.27	0.25	0.24	0.016	0.788

CK = dairy cows fed a basal diet (without flaxseed); WF = dairy cows with whole flaxseed diet (whole flaxseed 1500 g per day); GF = dairy cows with ground flaxseed diet (ground flaxseed 1500 g per day); ^a–c^ Means in the same row with different superscripts differ significantly for treatment effect.

**Table 7 animals-12-01898-t007:** Plasma parameters of dairy cows in different groups.

Item	Treatments			SEM	*p*-Value
	CK	WF	GF		
TG, mmol/L	1.17	1.00	1.07	0.071	0.615
LDLC, mmol/L	1.77	1.39	2.09	0.186	0.317
VLDLC, mmol/L	0.53	0.53	0.48	0.027	0.677
HDLC, mmol/L	1.43	1.48	1.30	0.053	0.379
TC, mmol/L	4.29	4.14	4.64	0.144	0.359
AST, U/L	17.81	16.13	14.4	1.688	0.726
ALT, U/L	20.08	20.74	25.87	1.979	0.440
ALP, U/L	97.03	109.96	86.57	5.889	0.310
TP, g/L	71.05	73.79	75.10	1.368	0.726
ALB, g/L	46.31	44.58	41.34	1.135	0.196
GLB, g/L	24.74	29.21	30.76	1.492	0.238
BUN, mmol/L	5.33	6.07	5.55	0.247	0.471
GLU, mmol/L	1.35	0.99	1.07	0.113	0.413

CK = dairy cows fed a basal diet (without flaxseed); WF = dairy cows with whole flaxseed diet (whole flaxseed 1500 g per day); GF = dairy cows with ground flaxseed diet (ground flaxseed 1500 g per day). TG = triglyceride; LDL = low-density lipoprotein cholesterol; VLDL = very low-density lipoprotein cholesterol; HDL = high-density lipoprotein cholesterol; TC = total cholesterol; Apo A = apolipoprotein A; Apo B = apolipoprotein B; AST = aspartate aminotransferase; ALT = alanine aminotransferase; ALP = alkaline phosphatase; TP = total protein; ALB = albumin; GLB = globulin; BUN = urea nitrogen; GLU = glucose.

## Data Availability

Not applicable.

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
