# Peer review of "Effect of Flaxseed Supplementation on Milk and Plasma Fatty Acid Composition and Plasma Parameters of Holstein Dairy Cows"

_animals, 2022, doi:10.3390/ani12151898_

Round 1

Reviewer 1 Report

Summary

According to the authors, the objective of this study was to explore the effect of whole and ground flaxseed supplementation on fatty acid composition (particularly n-3 PUFAs) in plasma and milk, milk production, and plasma parameters, in Holstein dairy cows.

The results indicated that compared with whole flaxseed supplementation, ground flaxseed supplementation showed higher efficiency in increasing total n-3 PUFA levels and milk ALA/ALA intake, and decreased the ratio of n-6 PUFAs to n-3 PUFAs in milk; causing no cardiovascular and liver harm to dairy cattle.

Major Issues

I think there is room for improvement in the discussion section (see comments attached in the manuscript itself), and that the last section needs to be rounded off with a more general conclusion.

Minor Issues

They are attached in the manuscript itself.

Author Response

On behalf of my co-authors, thank you very much for reviewing our manuscript. We appreciate the positive, constructive comments and suggestions on our manuscript entitled “Effect of Flaxseed Supplementation on Milk and Plasma Fatty Acid Composition and Plasma Parameters of Holstein Dairy Cows”.

The comments were valuable and helpful in revising and improving our paper and provided important guidance to our research. We studied the reviewers’ comments carefully and have made revisions based on these comments.

The main corrections and our point-by-point responses to the reviewers’ comments are as follows:

Others are modified in the article and marked with highlights.

Title

But where? In Holstein dairy cows? It should be mentioned.

[Response]

We had revised the title “Effect of Flaxseed Supplementation on Milk and Plasma Fatty Acid Composition and Plasma Parameters of Holstein Dairy Cows”.

2.2 Sampling, measurements, and analysis

They are not even mentioned in the rest of the article

[Response]

We have removed this part.

3.4 Milk fatty acid composition

Not so, that was in the plasma sample.

[Response]

We had revised “221% and 275%” to “223% and 350%” as showed in L174-175.

3.6 Correlation analysis

“Most of the n-3 PUFA concentration showed the negative correction with and n-6 PUFA in blood especially DGLA, GLA, AA and n-6 PUFA in plasma.” This sentence cannot be understood.

[Response]

We have revised this sentence as “Most of the n-3 PUFA concentration in milk showed the negative correction with and n-6 PUFA in plasma, especially DGLA, GLA, AA and n-6 PUFA in plasma.” as showed in L219-221.

4.2 Milk composition

“However, different forms of flaxseed may affect the protein in milk. Caroprese et al. [34] found that WF supplementation significantly increased the content of milk protein. Petit et al.[38] found 9.7% WF supplementation had no effect on milk protein content. Oba et al.[41] observed no difference between WF and GF supplementation at the same level. These results were similar to those of the present study, in which milk protein content differed between the CK and WF groups, but not between CK and GF. Moreover, compared with GF, WF supplementation non-significantly increased the milk protein content. This may be because, compared with GF, the WF showed higher efficiency in decreasing degradability of the protein in the rumen, and so increased protein in milk.” There are many contradictions or things that are not clear at all. This part needs a lot of improvement.

[Response]

We have revised this sentence as “However, different forms of flaxseed may affect the protein in milk. Caroprese et al. [34] found that WF supplementation significantly increased the content of milk protein. Is-enberg et al.[14] found GF supplementation had no effect on milk protein content. Oba et al.[41] observed no difference between WF and GF supplementation at the same level. These results were similar to those of the present study, in which milk protein content differed between the CK and WF groups, but not between CK and GF. Moreover, com-pared with GF, WF supplementation non-significantly increased the milk protein con-tent. This may be because, compared with GF, the WF showed higher efficiency in decreasing degradability of the protein in the rumen, and so increased protein in milk.” as showed in L267-275

4.5 Antioxidant indexes in plasma and milk

“Omega-3 polyunsaturated fatty acids showed high antioxidant activity [56], and increase activity of antioxidants in blood[57].” It is not at all clear what is meant by this phrase.

[Response]

We have revised the sentence as following:

As we know, n-3 PUFA showed high antioxidant activity[56], and diet n-3 PUFA can in-crease activity of antioxidants in blood[57], as showed in L342-343

“In this study, ground and whole flaxseed supplementation can increase the activity of GSH-Px in plasma,” This is not true, as the end of the sentence says ("no difference showed compared with CK group").

[Response]

We have revised the sentence as following:

In this study, ground and whole flaxseed supplementation showed trend to increase the activity of GSH-Px in plasma, as showed in L346-347.

Reviewer 2 Report

The manuscript “Effect of Flaxseed Supplementation on Milk and Plasma Fatty Acid Composition and Plasma Parameters” has been reviewed.

Generally, usage of oil seeds, including flaxseed, either intact or technologically processed for improvement of milk fatty acids profile is not new or original. The novelty of the manuscript is in providing information on the effect of oil seeds on plasma parameters, antioxidant indices and the correlations between n-3 and n-6 FA in milk and plasma, this information is lacking.

Overall, the manuscript is clear and well-written and requires only minor changes as specified bellow. Maybe, calculation of some other indices characterising health properties of milk fat could be usefull, e.g. atherogenic or trombogenic indices or hypocholesterolaemic/hypercholesterolaemic ratio.

 Specific comments - minor

Simple summary

L. 6 – PUFA (typing mistake)

Abstract

L. 7 – 4% energy-corrected milk – please replace by commonly used term 4% fat-corrected milk

Please add P-values to your results.

Material and Methods

P.4, L. 8 – fat-corrected milk

Results

Footnote of Tab 3 - DMI, dry matt – remove, 4%FCM, 4% fat corrected milk

Footnote of Tab. 4 and 5 – n-3 PUFA = ω-3 polyunsaturated fatty acid; n-6 PUFA = ω-6 polyunsaturated fatty acid (typing mistake)

Part 3.5, L. 6 - The concentration of (beginning of sentence)

Discussion

Part 4.1, L. 18 - This difference may be due to flaxseeds…

Part 4.4. L. 1-3 - A study by Brzozowska et al.[6] found supplemented cows with WF could decrease the fatty acids with a lower number of 18 carbons in milk, similar study also found in the study by Ward et al.[48]. – does not make sense, please reformulate

L. 7 - with a lower number of 18 carbons in milk - maybe better …than…

L. 13 - n-3 PUFAs

L. 23 - in vivo (in italics), but there are few focused on it – few what? studies?

P. 14, L. 4, L. 6 - positive correction??? probably positive correlation

P.15, Par. 2, L. 6 and 7 ­- This (beginning of the sentence) may can (may or can?) reduce the oxidative stress response of infant animals. … negatively correlated

Part 4.6 L. 9 - … in the present. Probably in the present study/work/experiment

References

Please check the reference style, mainly capital letters at the beginning of the Journal titles (Animal feed science and technology x Animal Feed Science and Technology)

Author Response

On behalf of my co-authors, thank you very much for reviewing our manuscript. We appreciate the positive, constructive comments and suggestions on our manuscript entitled “Effect of Flaxseed Supplementation on Milk and Plasma Fatty Acid Composition and Plasma Parameters of Holstein Dairy Cows”.

The comments were valuable and helpful in revising and improving our paper and provided important guidance to our research. We studied the reviewers’ comments carefully and have made revisions based on these comments.

The main corrections and our point-by-point responses to the reviewers’ comments are as follows:

Others are modified in the article and marked with highlights

L.6 – PUFA (typing mistake)

[Response]

We have revised “UPFA” to “PUFA” as showed in L22.

Abstract

  1. 7 – 4% energy-corrected milk – please replace by commonly used term 4% fat-corrected milk

[Response]

We have revised “4% energy-corrected milk” to “4% fat corrected milk” as showed in L29-30.

Please add P-values to your results.

[Response]

We have added P-values in Abstract part.

Material and Methods

P.4, L. 8 – fat-corrected milk

[Response]

We have revised “energy-corrected milk” to “fat corrected milk” as showed in L99.

Results

Footnote of Tab 3 - DMI, dry matt – remove, 4%FCM, 4% fat corrected milk

We have removed “DMI, dry matt” and revised “energy-corrected milk” to “fat corrected milk” as showed in L143.

Footnote of Tab. 4 and 5 – n-3 PUFA = ω-3 polyunsaturated fatty acid; n-6 PUFA = ω-6 polyunsaturated fatty acid (typing mistake)

[Response]

We have removed “n-3 PUFA = ω-3 polyunsaturated fatty acid; n-6 PUFA = ω-6 polyunsaturated fatty acid”.

Part 3.5, L. 6 - The concentration of (beginning of sentence)

[Response]

We had revised “the” to “The”, as showed in L203.

Discussion

Part 4.1, L. 18 - This difference may be due to flaxseeds…

[Response]

We had revised “different” to “difference” and showed in L246.

Part 4.4. L. 1-3 - A study by Brzozowska et al.[6] found supplemented cows with WF could decrease the fatty acids with a lower number of 18 carbons in milk, similar study also found in the study by Ward et al.[48]. – does not make sense, please reformulate

[Response]

We have revised this sentence “Brzozowska et al.[6] and Ward et al.[48] reported that WF supplemented in diet of cows could decrease the fatty acids with a lower number of 18 carbons in milk.”, as showed in L297-299.

  1. 7 - with a lower number of 18 carbons in milk - maybe better …than…

[Response]

We had revised showed in L303

  1. 13 - n-3 PUFAs

[Response]

We had revised “n-3 UPFAs” to “n-3 PUFAs”, as showed in L309.

  1. 23 - in vivo (in italics), but there are few focused on it – few what? studies?

[Response]

Few studies focused on it, as showed in L319.

  1. 14, L. 4, L. 6 - positive correction??? probably positive correlation

[Response]

We had revised “positive correction” to “probably positive correlation”.

P.15, Par. 2, L. 6 and 7 ­- This (beginning of the sentence) may can (may or can?) reduce the oxidative stress response of infant animals. … negatively correlated

[Response]

We had revised to “reported that pregnant women eating n-3 PUFAs food may can reduce the oxidative response of stress in infants”, as showed in L352-353.

Part 4.6 L. 9 - … in the present. Probably in the present study/work/experiment

[Response]

We had revised to “in the present study”, as showed in L367.

References

Please check the reference style, mainly capital letters at the beginning of the Journal titles (Animal feed science and technology x Animal Feed Science and Technology)

[Response]

We had revised the all reference styles.

Reviewer 3 Report

Overall comment: this study evaluated the effect of whole flaxseed and ground flaxseed supplementation on the composition of fatty acids in plasma and milk, particularly content of n-3 PUFAs. An interesting set of plasma metabolites has been evaluated. The study is interesting, concise and reasonably conducted.

Few minor suggestions and comments are written below.

- lines 5-6: "4" should be superscript

- Introduction: the authors chose to measure blood parameters to check the health status of the cows with changes of diets. Milk fatty acid profile is known to be a fingerprint of health status of the cow. Please add few lines on this and the reported associations between changing in milk fatty acid profile and blood parameters (so, not only human with milk  but also cows' health monitoring during changing in diet supplementation). Please see PMID: 35565628, PMID: 15736916, PMID: 27665132.

- line 131: please include in the table's caption the acronyms of treatment ( or in the footnotes) tables and figures should be self-explanatory.

- Discussion section: the significance of the lack of effects of different diets on plasma parameters should be shortly expanded (lines 366-369) using appropriate references as a comparison. Indeed, should be mentioned (with few lines discussion in the paragraph 4.4) that changes in milk fatty acid profile (that resulted using different diets' supplementation) could be associated to variation on blood parameters, with putative significance on health status of the individual. For instance, a recent study has been published on the associations between milk and fatty acid profile and blood parameters in dairy cows (PMID: 35565628). In addition, less recent but important studies underlined these relationship (PMID: 25200787 and PMID: 26094221).

IMO, this information should be included in the discussion to strengthen the significance of your results, especially in relation to oxidative stress parameters on one hand, and changes on milk FA profile due to different diets' supplementation, on the other hand.

Author Response

Overall comment: this study evaluated the effect of whole flaxseed and ground flaxseed supplementation on the composition of fatty acids in plasma and milk, particularly content of n-3 PUFAs. An interesting set of plasma metabolites has been evaluated. The study is interesting, concise and reasonably conducted.

Few minor suggestions and comments are written below.

- lines 5-6: "4" should be superscript

[Response]

We had revised in the article.

- Introduction: the authors chose to measure blood parameters to check the health status of the cows with changes of diets. Milk fatty acid profile is known to be a fingerprint of health status of the cow. Please add few lines on this and the reported associations between changing in milk fatty acid profile and blood parameters (so, not only human with milk but also cows' health monitoring during changing in diet supplementation). Please see PMID: 35565628, PMID: 15736916PMID: 27665132.

[Response]

We had revised in the article, as showed in L69-71.

- line 131: please include in the table's caption the acronyms of treatment (or in the footnotes) tables and figures should be self-explanatory.

[Response]

We had added in the article.

- Discussion section: the significance of the lack of effects of different diets on plasma parameters should be shortly expanded (lines 366-369) using appropriate references as a comparison. Indeed, should be mentioned (with few lines discussion in the paragraph 4.4) that changes in milk fatty acid profile (that resulted using different diets' supplementation) could be associated to variation on blood parameters, with putative significance on health status of the individual. For instance, a recent study has been published on the associations between milk and fatty acid profile and blood parameters in dairy cows (PMID: 35565628). In addition, less recent but important studies underlined these relationship (PMID: 25200787 and PMID: 26094221).

[Response]

We had revised in the article, as showed in L379-380.

IMO, this information should be included in the discussion to strengthen the significance of your results, especially in relation to oxidative stress parameters on one hand, and changes on milk FA profile due to different diets' supplementation, on the other hand.

[Response]

We had revised in the article, as showed in L394-398.

Reviewer 4 Report

The present manuscript describes the effect of different dietary sources of flaxseeds on thecow milk quality (composition of FA mainly) and selected blood indicators of dairy cows. In my opinion, the study issue has interest from scientific and research point of view. The experiment design and the aims seem to be good. However, there are some flaws in the present version of manuscript that could be revised/reworked and also some questions could be clarified clearly. I've made specific comments/suggestions directly in the attached pdf file (in the original submitted pdf version of the manuscript).

Author Response

L18-19 “milk α-linolenic acid (ALA) convert efficiency (milk ALA/ALA intake)” Please, rewrite this part when use better EN stylistics.

[Response]

We had revised to “efficiency of diet α-linolenic acid (ALA) convert to milk (milk ALA/ALA intake ratio)” As showed in L21-22.

L37-39 “This result indicated that compared with WF, GF supplementation showed higher efficiency in increasing total n-3 PUFA levels and milk ALA/ALA intake, and decreased the ratio of n-6 PUFAs to n-3 PUFAs in milk.” Please, rewrite this sentence to be better understandable for readers of the journal.

[Response]

We had revised to “This result indicated that compared with WF group, GF supplementation in diet showed higher efficiency in increasing total n-3 PUFA levels and milk ALA/ALA intake ratio, and decreased the ratio of n-6 PUFAs to n-3 PUFAs in milk.)” As showed in L40-42.

L81 Please add the following specifications:

the source of the flaxseeds used in the diet – event. variety of the flax; the length of the study (experimental feeding and evaluation); the parity and lactation stage of cows included in the study.

[Response]

Due to the high repetition rate of the article, this part is not described in this article, please see our previous published paper.

(Huang, G.X., Guo, L.Y., Chang, X.F., Liu, K.Z., Tang, W.H., Zheng, N., Zhao, S.G., Zhang Y.D., Wang J.Q. Effect of Whole or Ground Flaxseed Supplementation on Fatty Acid Profile, Fermentation, and Bacterial Composition in Rumen of Dairy Cows. Front Microbiol 2021, 12, 760528.),

L269-271 “This may be because, compared with GF, the WF showed higher efficiency in decreasing degradability of the protein in the rumen, and so increased protein in milk.” Please, rewrite this sentence when you use better EN stylistics.

[Response]

We had revised to “This may due to compared with GF, the WF showed higher efficiency in decreasing degradability of the protein in the rumen, and so increased protein in milk.” As showed in L291-293.

L342-346 “In this study, ground and whole flaxseed supplementation showed trend to increase the activity of GSH-Px in plasma, but no different showed compared with CK group, this result was similar with study by Pi et al [10]. Malondialdehyde is the main product of lipid peroxidation, in this experiment no different show between the treatments.” Please, correct the EN grammar and stylistics.

[Response]

We had revised to “This study showed that dietary supplementation of whole and ground flaxseed increased the activity of GSH-Px in plasma. but did not differ from that of the control group.” As showed in L364-366.

L348-351 Prescott et al.[58] reported that pregnant women eating n-3 PUFAs food may can reduce the oxidative response of stress in infants. Similar result was also found in this experiment, ground and whole flaxseed supplementation can increase the activity of GSH-Px, and decrease the concentration of MDA in milk.

[Response]

We had revised to “Prescott et al. [61] found that pregnant women consumption of rich in n-3 PUFAs may reduce the oxidative stress in infants. Similar results were found in this study, supplementation with whole and ground flaxseed contributed to the activity of GSH-Px and reduced the content of MDA in milk.” As showed in L368-371.

Reviewer 5 Report

Abstract

The use of so many abbreviations makes reading of the abstract difficult. Could the authors please rephrase to make flow of text easier?

Introduction

Please define clearly the objectives of the study.

Materials and methods

2.1. Please describe clearly and in detail, the criteria for allocation of animals into the three groups.

2.3. Please provide proof that results had a normal distribution, or, otherwise please use non-parametric tests for analyses.

Results

Tables 3 and 4 are too detailed, please move to supplementary material.

Figure 1 is located away from citing text.

Discussion

Please add a subsection 4.7 to discuss together all the preceding findings.

Also, many relevant references from other researchers are missing, please add.

Overall

Please make corrections and resubmit improved version.

Author Response

Abstract

The use of so many abbreviations makes reading of the abstract difficult. Could the authors please rephrase to make flow of text easier?

[Response]

We had revised showed in L27-28 and L33-34.

Introduction

Please define clearly the objectives of the study.

[Response]

We had revised showed in L75-77.

Materials and methods

2.1. Please describe clearly and in detail, the criteria for allocation of animals into the three groups.

[Response]

Due to the high repetition rate of the article, this part is not described in this article, please see our previous published paper.

(Huang, G.X., Guo, L.Y., Chang, X.F., Liu, K.Z., Tang, W.H., Zheng, N., Zhao, S.G., Zhang Y.D., Wang J.Q. Effect of Whole or Ground Flaxseed Supplementation on Fatty Acid Profile, Fermentation, and Bacterial Composition in Rumen of Dairy Cows. Front Microbiol 2021, 12, 760528.),

2.3. Please provide proof that results had a normal distribution, or, otherwise please use non-parametric tests for analyses.

[Response]

The data in present studies was tested by the HOVTEST and UNIVARIATE of the SAS.

Results

Tables 3 and 4 are too detailed, please move to supplementary material.

[Response]

Tables 3 and 4 are main result of our article, thus we recommend to keep them in the text

Figure 1 is located away from citing text.

[Response]

We had revised in article.

Discussion

Please add a subsection 4.7 to discuss together all the preceding findings.                                                                                                                                       

[Response]

We had added in article, as showed in L394-398.

Also, many relevant references from other researchers are missing, please add.

[Response]

We had revised in article.

Overall

Please make corrections and resubmit improved version.

Round 2

Reviewer 1 Report

I don't see that the article has improved much; indeed, in some respects it has even worsened.

In addition to all the new comments attached in the modified manuscript itself, where are the modifications corresponding to “the last section needs to be rounded off with a more general conclusion”, which I described as a Major Issue????.

Apart from that, comments I made about Table 1 have also not been taken into account in what is now Supplementary Table1.

Author Response

Table 4. “SFA = Saturated fatty acids; MUFA = Monounsaturated fatty acids” These should be defined in Table 3, not in Table 4.

[Response]

We had deleted it in table 4.

“Most of the n-3 PUFA concentration in milk showed the negative correction with and n-6 PUFA in plasma, especially DGLA, GLA, AA and n-6 PUFA in plasma.” is difficult to understand.

[Response]

We had revised this sentence as showed in L237-238.

L263-L271 The authors have a very peculiar view on the meaning of "needing many improvements". I do not see more than a small improvement.

[Response]

We had revised this sentence as showed in L287-294.

L324 and L326 Probably???? What do you use statistics for then?

[Response]

We had deleted it in this article.

I say it again: this expression seems to me to be incorrect. Moreover, it seems to me that the reader will not understand what is meant.

And I say it again: What is this "activity of antioxidants level ????????

[Response]

We had revised “activity of antioxidants level” to “activity of antioxidants”. As showed in L364, L365, L369, L371, and L372-373

L531 Who said to remove the sentence that was here????

It is our mistaken, and we have added “This may can reduce the oxidative stress response of infant animals.” in article, as showed in L373-374.

“the oxidative response of stress in infants.” I disagree, that's not what I said in my comment

[Response]

We had revised to “the oxidative stress in infants.” As showed in L371.

Reviewer 5 Report

2.1. Please describe clearly and in detail, the criteria for allocation of animals into the three groups.

[Response]

Due to the high repetition rate of the article, this part is not described in this article, please see our previous published paper.

(Huang, G.X., Guo, L.Y., Chang, X.F., Liu, K.Z., Tang, W.H., Zheng, N., Zhao, S.G., Zhang Y.D., Wang J.Q. Effect of Whole or Ground Flaxseed Supplementation on Fatty Acid Profile, Fermentation, and Bacterial Composition in Rumen of Dairy Cows. Front Microbiol 2021, 12, 760528.),

I still suggest STRONGLY  to include criteria for allocation of animals into groups in the revised manuscript. The readers will not wish to refer to another article when reading this publication. Also, the reluctance of the authors may raise suspicions that the procedure was not correct. I suggest to rectify the problem.

Author Response

I still suggest STRONGLY  to include criteria for allocation of animals into groups in the revised manuscript. The readers will not wish to refer to another article when reading this publication. Also, the reluctance of the authors may raise suspicions that the procedure was not correct. I suggest to rectify the problem.

[Response]

Thank you for your suggestion, and we added the criteria for allocation of animals in article “(days in milk: 90 ± 28 d; body weight: 628 ± 103 kg; milk yield: 37.22 ± 2.60 kg)” as showed in L81-82.

Round 3

Reviewer 1 Report

For the second time, I still do not see any modification or response corresponding to my comment “the last section needs to be rounded off with a more general conclusion”, which I described as a Major Issue.

Also for the second time, comments I made about Table 1 have also not been taken into account in what is now Supplementary Table1.

I have included two new comments in your response file (attached here).

Author Response

Q1. But you have not added it in table 3, so it is undefined.

[Response]

We are frightfully sorry, It is our mistaken, we have removed in all table.

Q2. I have already said that I did not agree with this sentence. What I said was: "This can reduce the oxidative stress in infant animals"

[Response]

Thanks for your suggestion, and we have revised in this article as showed in L376-377.
